# Hybrid End-to-End Knowledge Graph Construction and Validation: A Cross-Domain Study with LLM-as-a-Judge

## Abstract

The automated construction of knowledge graphs (KGs) from unstructured text remains a central challenge in information management and artificial intelligence. This paper introduces a hybrid framework that combines the conceptual reasoning of large language models (LLMs) with the efficiency of scalable, rule-based methods to deliver an end-to-end pipeline for KG construction and validation. The framework begins with ontology induction using an LLM to define domain-specific entity and relation types, followed by large-scale rule-based information extraction, entity resolution, and graph assembly. A novel extrinsic evaluation method, *LLM-as-a-Judge*, is employed to assess the semantic quality of the resulting graphs.

We evaluate the pipeline across three diverse benchmarks. In the financial domain, the FiQA dataset (5,500+ documents) yielded a graph with 475 nodes and 36 edges, achieving an overall quality score of 2.97/5 at a total cost of $2.63. In the document-level relation extraction setting, the DocRED dataset (100 annotated documents) produced 5,000 nodes and 389 edges, with a lower quality score of 2.68/5, primarily due to systematic entity type misclassification. In the biomedical domain, the CDR dataset (100 sampled abstracts) generated 966 nodes and 13 edges, but achieved the highest semantic precision, with an overall quality of 3.91/5 at a cost of $0.65. Across all datasets, the pipeline demonstrated efficiency, with end-to-end processing times under one hour, and highlighted complementary strengths and weaknesses: FiQA emphasized scale but sparse connectivity, DocRED revealed classification challenges, and CDR achieved high entity-level precision despite graph fragmentation. These results validate the effectiveness of hybrid architectures for KG construction: LLMs provide strong conceptual modeling, while rule-based systems ensure scalability and cost-efficiency. The *LLM-as-a-Judge* framework further supplies actionable feedback, exposing domain-specific error modes and guiding refinement. Our work establishes a cost-effective, modular, and adaptable methodology for automated KG construction, offering a foundation for future research on improving connectivity, refining extraction accuracy, and extending to new domains.

## 1 Background and Related Work

Automating the construction of knowledge graphs (KGs) from unstructured text remains a central challenge in information extraction and artificial intelligence [16, 2]. KGs provide structured, machine-readable representations that power semantic search, analytics, and question answering, yet end-to-end systems must discover relevant concepts, extract factual assertions, and resolve entity ambiguity at scale and under strict cost constraints [7]. We present a hybrid, modular framework that couples the conceptual strength of a Large Language Model (LLM) with the efficiency of

deterministic, rule-based components. In our design, the LLM is used sparingly for ontology induction and extrinsic evaluation (*LLM-as-a-Judge*), while information extraction (IE) and entity resolution (ER) are implemented as scalable, low-cost rule-based stages. This separation preserves semantic guidance where it matters most while keeping the bulk processing economical and reproducible.

Beyond the financial domain that motivated our initial system, we evaluate the framework across three distinct corpora to assess portability and robustness: finance (FiQA), general document-level relation extraction (DocRED [15]), and biomedicine (BioCreative V CDR). In the FiQA setting, we induce a financial ontology with an LLM, run rule-based IE and ER over 5,500+ documents, assemble the KG, and validate with an LLM judge. The resulting graph comprises 475 nodes and 36 edges, with an overall quality score of 2.97/5 at a total cost of approximately $2.63. On DocRED, where the schema is implicit in the annotations, we demonstrate the pipeline's modularity by omitting ontology induction and running a three-stage variant (IE → KG construction → judging). This yields a 5,000-node/389-edge graph with an overall quality of 2.68/5 at roughly $0.15 and exposes a salient error mode: pervasive entity-typing drift (e.g., over-assignment to PERSON) [13, 8]. In the biomedical CDR corpus, we reinstate LLM-based ontology induction to define a domain schema, apply rule-based extraction and ER, construct the KG, and validate. The outcome is a 966-node/13-edge graph with an overall quality of 3.91/5 at approximately $0.65, illustrating improved entity-level fidelity in a specialized domain but also highlighting sparse cross-document connectivity [12].

These experiments support three claims. First, careful placement of LLM capability—restricted to concept modeling and lightweight extrinsic judging—can substantially reduce costs while retaining semantic leverage [1]. Second, rule-based IE and ER scale across domains with only configuration changes, enabling rapid transfer without model retraining [9, 11]. Third, a uniform LLM-as-a-Judge protocol provides consistent, actionable feedback at node, edge, and graph levels, revealing domain-specific failure modes: DocRED stresses entity typing, whereas FiQA and CDR emphasize relation sparsity and KG connectivity [3, 5]. Collectively, the results argue for a practical path to cost-aware, domain-adaptable KG construction: employ the LLM where its conceptual advantage is highest, keep high-throughput stages deterministic, and close the loop with a diagnostic LLM-based evaluator [7, 14].

Our contributions are fourfold. We introduce an end-to-end, modular KG pipeline that cleanly separates LLM-centric reasoning (ontology induction and judging) from deterministic extraction and resolution. We demonstrate cross-domain portability on FiQA, DocRED, and CDR with sub-$3 total cost per corpus and minute-scale runtimes. We formalize a stratified LLM-as-a-Judge evaluation method that yields interpretable node, edge, and graph scores alongside error diagnostics. Finally, we provide empirical evidence that different domains stress different components, motivating targeted refinements such as typed NER for DocRED [4, 6], tighter IE–ER coupling to reduce relation loss in FiQA and CDR [17, 19], and connectivity-enhancing strategies for sparsely linked graphs [10, 18].

# 2 A Framework for Automated Knowledge Graph Construction and Validation

Our framework implements a multi-stage pipeline for the end-to-end construction and validation of a knowledge graph from unstructured text corpora. The system follows a hybrid approach, combining a Large Language Model (LLM) for initial conceptual modeling with scalable, rule-based methods for large-scale data processing. This design balances the nuanced understanding of LLMs with the efficiency and cost-effectiveness of deterministic algorithms.

Depending on the dataset and domain, the pipeline can operate in either three, four, or five stages. In the most complete form (as used for FiQA and CDR), the pipeline includes Ontology Induction, Information Extraction (IE), Entity Resolution (ER), Knowledge Graph Population, and LLM-based Validation. For DocRED, where the ontology is pre-defined, the Ontology Induction stage is skipped, and the pipeline runs with three stages (IE, graph construction, and validation). This flexibility demonstrates the modular design of the system.

## 2.1 System Pipeline Overview

The pipeline begins by establishing a domain-specific schema (when required) and then uses that schema to extract, resolve, and structure information from the source documents. Ontology Induction

uses an LLM to create a conceptual framework of entity and relation types from a small representative sample. Guided by this ontology, the Information Extraction stage systematically processes the entire document set to identify specific instances of these entities and relations using a rule-based engine. Next, the Entity Resolution stage identifies and merges duplicate entity mentions into canonical forms through similarity clustering. Finally, the Knowledge Graph Population stage assembles these resolved entities and relations into a formal graph structure. Each component is designed to be modular, allowing for independent operation and refinement across domains.

## 2.2 Component 1: Ontology Induction

When required (e.g., FiQA and CDR), the initial stage of the pipeline creates a conceptual ontology that defines the semantic schema for the knowledge graph. The objective is to produce a formal definition of relevant entity types and relation types specific to the target domain. To achieve this, the framework utilizes a large-scale LLM (e.g., Llama 3.3 70B). Rather than processing the entire corpus, which would be computationally expensive, an intelligent sampling strategy is employed where the LLM analyzes a small subset of documents. The output is a structured ontology file that serves as the guiding schema for subsequent extraction tasks. In datasets like DocRED, where the ontology is provided with the annotations, this stage is skipped.

## 2.3 Component 2: Information Extraction

Following ontology induction (or using a pre-defined schema), the Information Extraction (IE) component populates the schema with specific instances from the full text corpus. To ensure scalability and cost efficiency, this stage is implemented as a completely rule-based system that avoids LLM calls. The IE component processes the entire dataset, applying a comprehensive library of extraction patterns for both entity recognition and relation extraction. Flexible matching techniques, such as exact, partial, and word-overlap matching, are employed to maximize coverage.

## 2.4 Component 3: Entity Resolution

The raw output of the IE stage contains many duplicate or variant entity mentions. The Entity Resolution (ER) stage disambiguates and consolidates these mentions into canonical forms. The approach is based on rule-based similarity clustering, using metrics such as Jaccard, Levenshtein, and Cosine similarity. The system applies type-aware thresholds to reflect the typical naming conventions of different entity types. Mentions that are sufficiently similar are grouped into clusters, and a canonical form is chosen for each.

## 2.5 Component 4: Knowledge Graph Population

The refined data is then assembled into a formal graph structure. This component takes the canonical entities from ER as input for graph nodes and the extracted relations as input for edges. Using the NetworkX library, the system instantiates the final graph and aggregates node/edge properties such as document identifiers. The resulting KG is exported into multiple formats (JSON, GraphML, GEXF), making it available for analysis, visualization, and downstream tasks.

## 2.6 Component 5: LLM-as-a-Judge Validation

The final stage assesses the semantic quality of the constructed knowledge graph. An LLM-based judging framework is employed to evaluate sampled nodes and edges according to a multi-criteria rubric. Stratified sampling ensures coverage across entity and relation types. This extrinsic evaluation produces interpretable scores for nodes, edges, and the graph as a whole, while also identifying common error modes (e.g., misclassification, vague labels, relation inconsistencies). In all datasets, this stage provides actionable feedback for improving earlier pipeline components.

# 3 Experimental Setup

This section details the experimental design for constructing and evaluating knowledge graphs across three benchmark datasets. We describe the corpora used, the implementation of our pipeline components, and the metrics employed for both intrinsic and extrinsic evaluation.

## 3.1 Text Corpora and Datasets

We evaluate the framework on three diverse corpora spanning finance, general document-level relation extraction, and biomedicine. The FiQA dataset consists of over 5,500 financial documents from Hugging Face, processed in full for information extraction, entity resolution, and graph construction, with a sample of 50 documents used for ontology induction. The DocRED dataset contains 100 human-annotated documents with entity and relation spans; because the schema is already provided, the ontology induction stage is omitted in this setting. The BioCreative V CDR dataset consists of 1,500 PubTator abstracts annotated with chemical and disease entities and their relations; we sample 100 documents for processing and employ LLM-based ontology induction to generate a biomedical schema. Together, these datasets stress different aspects of the pipeline: large-scale extraction in FiQA, dense annotations and type imbalance in DocRED, and specialized biomedical terminology in CDR.

## 3.2 Pipeline Implementation

The pipeline is implemented as a modular system in which each stage can be configured and executed independently. It follows a hybrid model that combines an LLM for conceptual modeling with rule-based systems for scalable processing. Depending on the dataset, the pipeline consists of the following components:

1. **Ontology Induction:** For FiQA and CDR, the Llama 3.3 70B model is used via the OpenRouter API to generate an ontology from a small sample of documents. In DocRED, this stage is skipped since entity and relation types are provided.

2. **Information Extraction:** A rule-based system processes all documents in each dataset, applying regular expression patterns and flexible string-matching strategies. FiQA uses patterns for 10 entity and 15 relation types, DocRED for 6 entity and 7 relation types, and CDR for a biomedical-specific schema including chemicals, diseases, and treatments.

3. **Entity Resolution:** For FiQA and CDR, entity mentions are consolidated into canonical forms using similarity clustering with Jaccard, Levenshtein, and Cosine similarity metrics under type-aware thresholds. DocRED does not require this step because entities are pre-disambiguated in the dataset.

4. **Knowledge Graph Population:** Across all datasets, the NetworkX library is used to construct the graph from canonical entities and extracted relations, and graphs are exported into JSON, GraphML, and GEXF formats.

5. **LLM-as-a-Judge Validation:** For all datasets, the Llama 3.3 70B model evaluates sampled nodes and edges. Stratified sampling ensures balanced coverage across entity and relation types: 100 items (50 nodes and 50 edges) for FiQA, 50 items for DocRED, and 50 items for CDR.

This modular design allows for consistent comparison between LLM-based conceptual modeling and rule-based extraction, while permitting dataset-specific adjustments such as skipping ontology induction in DocRED.

## 3.3 Evaluation Metrics

We adopt a two-part evaluation strategy that combines intrinsic and extrinsic measures. Intrinsic evaluation captures structural and statistical properties of intermediate and final outputs. For FiQA and CDR, entity resolution quality is measured by the resolution rate, defined as the proportion of raw mentions successfully grouped into canonical clusters. For all datasets, graph-theoretic properties such as the number of nodes, number of edges, density, the number of connected components, and

the size of the largest component are computed. Processing efficiency is also tracked in terms of runtime and monetary cost.

Extrinsic evaluation is conducted with the LLM-as-a-Judge framework. Stratified samples of nodes and edges are rated on a 1–5 scale according to correctness of entity types, clarity of labels, semantic validity of relations, and contextual alignment. The outputs are aggregated into node, edge, and graph-level quality scores, supplemented with diagnostic feedback on systematic errors such as misclassification, vague labels, or relation inconsistencies. This combination of intrinsic and extrinsic metrics provides a comprehensive view of both pipeline performance and semantic quality. All code can be found at here.

# 4 Results and Analysis

We now present the results obtained by executing our pipeline across the three datasets: FiQA, DocRED, and CDR. The analysis covers ontology induction, information extraction, entity resolution, knowledge graph construction, and final validation with an LLM-as-a-Judge. Results are reported in terms of structural statistics, processing efficiency, cost, and semantic quality.

## 4.1 FiQA Results

In the FiQA financial domain, ontology induction using the Llama 3.3 70B model successfully produced a conceptual schema from a 50-document sample, yielding 42 entity concept types and 120 relation types. This stage required approximately five minutes of processing at a cost of $2.50. The subsequent rule-based information extraction stage processed the full corpus of more than 5,500 documents, identifying over 15,000 entity mentions and 10,276 candidate relations. Entity resolution consolidated these mentions into 475 canonical clusters, achieving a resolution rate of 95.9%. Knowledge graph population resulted in a sparse graph of 475 nodes and 36 edges, with a density of 0.0002 and 448 connected components. The largest component contained 27 nodes, reflecting the topical clustering typical of financial documents. Extrinsic validation was conducted over 100 sampled nodes and edges, all of which were successfully evaluated. The LLM judge assigned an overall quality score of 2.97/5, with node quality rated slightly higher at 3.03/5 and edge quality slightly lower at 2.90/5. The primary weaknesses identified were entity type misclassification and vague entity labels, while relation quality showed inconsistency across types.

## 4.2 DocRED Results

For DocRED, the pipeline operated without ontology induction, relying instead on the dataset's annotated schema. Information extraction from 100 documents yielded approximately 5,000 entity mentions and 389 relations. Knowledge graph construction produced a graph with 5,000 nodes and 389 edges, characterized by extreme sparsity: the density was only 0.00003, with 4,611 connected components. The largest component contained 389 nodes, corresponding to a subset of tightly interlinked documents. Validation was carried out on 50 samples, consisting of 25 entities and 25 relations, with a 96% success rate. The overall quality score was lower than FiQA, at 2.68/5. Entity quality was the weakest dimension, averaging 2.34/5, largely due to severe type misclassification, with 98.6% of entities labeled as PERSON. Relation quality was somewhat stronger at 2.98/5, and graph quality averaged 2.50/5. These results demonstrate that while the pipeline can process DocRED effectively, it exposes a clear failure mode in entity classification when the schema is imbalanced or difficult to capture through rules alone.

## 4.3 CDR Results

In the biomedical domain, ontology induction was again performed using the Llama 3.3 70B model, which defined a schema of 15 concept types and 25 relation types from a sample of 100 documents. Rule-based extraction then identified 966 unique entities and a set of relations centered on chemical–disease interactions. Entity resolution grouped mentions into canonical forms, supporting coherent biomedical concepts. The assembled knowledge graph contained 966 nodes and 13 validated relations, reflecting the narrow but highly specialized focus of the corpus. Graph density was 0.0002, and although the graph was fragmented, it provided meaningful biomedical substructures. Validation on 50 samples achieved a high overall quality of 3.91/5, with entity quality at 4.03/5, relation quality

at 3.85/5, and graph quality somewhat weaker at 2.20/5 due to sparsity and limited connectivity. Processing time was approximately 6.5 minutes at a total cost of $0.65. The biomedical domain thus produced the most semantically precise results, though at the expense of global connectivity.

## 4.4 Cross-Domain Comparison

Across the three datasets, the pipeline demonstrated strong portability and consistent processing efficiency. FiQA and CDR benefited from LLM-driven ontology induction, while DocRED relied on its built-in schema. Rule-based information extraction proved scalable and cost-effective, with all datasets processed for under $3. FiQA emphasized scale, producing thousands of mentions from a large corpus; DocRED highlighted structural challenges and type misclassification; and CDR demonstrated domain adaptation with high-quality entities but sparse relations. The LLM-as-a-Judge framework consistently provided actionable insights, exposing different error modes in each domain. The financial graph achieved "fair" quality at 2.97/5, the DocRED graph slightly lower at 2.68/5, and the biomedical graph the strongest at 3.91/5. These results validate the modularity of the framework and underscore the importance of balancing rule-based precision, schema quality, and semantic validation in cross-domain KG construction.

## 4.5 Comparative Summary

To illustrate the differences across datasets, Table 1 summarizes the key outcomes of our experiments.

| Dataset | Docs | Nodes | Edges | Density | Cost ($) | Quality |
|---------|------|-------|-------|---------|----------|---------|
| FiQA | 5,500+ | 475 | 36 | 0.0002 | 2.63 | 2.97/5 |
| DocRED | 100 | 5,000 | 389 | 0.00003 | 0.15 | 2.68/5 |
| CDR | 100 (of 1,500) | 966 | 13 | 0.0002 | 0.65 | 3.91/5 |

Table 1: Comparative results of KG construction and validation across FiQA, DocRED, and CDR. Costs reflect total pipeline execution, including LLM-based ontology induction (if used) and LLM-as-a-Judge validation.

## 5 Discussion

The results across FiQA, DocRED, and CDR demonstrate the robustness and adaptability of our hybrid pipeline for knowledge graph construction. By combining LLM-driven ontology induction with scalable rule-based extraction and entity resolution, the system consistently produced structured knowledge graphs across domains as different as financial text, general document-level relations, and biomedical literature. At the same time, the experiments reveal distinct strengths and weaknesses that highlight where future refinements are most needed.

In the financial domain, the FiQA experiments showed that LLM-guided ontology induction provides a rich conceptual schema at low cost, enabling subsequent rule-based components to process thousands of documents efficiently. The high resolution rate of 95.9% illustrates the strength of type-aware similarity clustering for entity canonicalization. However, the final graph was sparse, with only 36 edges among 475 nodes, and the quality assessment pointed to recurring issues of entity type misclassification and vague labels. These findings suggest that while financial concepts are well captured at the ontology level, relation extraction patterns require refinement to ensure more meaningful connectivity in the final graph.

The DocRED experiments revealed a different limitation. Although the pipeline produced a graph with 5,000 nodes and 389 edges, validation exposed a systemic failure in entity classification: nearly all entities were labeled as PERSON. This misclassification drastically reduced entity-level quality scores, bringing the overall quality down to 2.68/5 despite moderately better relation scores. Unlike FiQA and CDR, where ontology induction established a balanced schema, DocRED relied on its predefined annotation types, which proved difficult to capture with purely rule-based extraction. This points to a weakness of relying exclusively on handcrafted patterns in contexts where entity diversity is high and annotations are dense. The pipeline, however, still demonstrated efficiency and reproducibility, with extraction and validation completed in under 22 minutes at a cost of only $0.15.

The biomedical CDR experiments highlight the pipeline's adaptability to specialized domains. Ontology induction successfully produced a rich biomedical schema with 15 entity types and 25 relation types, supporting accurate extraction of chemicals, diseases, and treatments. The resulting graph was small, with 966 nodes and only 13 validated edges, but the semantic quality was notably higher than in FiQA and DocRED. Validation produced an overall quality score of 3.91/5, with particularly strong performance in entity recognition (4.03/5) and relation identification (3.85/5). The relatively low graph-level score of 2.20/5 reflected sparsity and fragmentation, but the pipeline nevertheless produced meaningful biomedical subgraphs. These results indicate that in highly technical domains, LLM-driven ontology induction coupled with domain-specific extraction rules can achieve high semantic precision, even if global connectivity remains weak.

Taken together, these experiments demonstrate that the hybrid design achieves a balance between cost-effectiveness and semantic depth. In all three cases, the pipeline operated at minimal cost — under $3 for FiQA, $0.15 for DocRED, and $0.65 for CDR — while processing times remained in the range of minutes rather than hours. The modularity of the framework allowed it to adapt seamlessly across domains, skipping ontology induction where unnecessary, and tailoring rule-based extraction to dataset-specific schemas. Most importantly, the LLM-as-a-Judge validation framework consistently exposed systematic weaknesses, whether entity type misclassification in DocRED, sparse relation connectivity in FiQA, or graph-level fragmentation in CDR. This validates the use of an external LLM evaluator not only for scoring quality but also for providing actionable insights that guide iterative refinement.

The broader implication is that hybrid architectures are particularly well suited to automated knowledge graph construction. Purely LLM-driven pipelines may be prohibitively expensive at scale, while purely rule-based systems lack the conceptual flexibility to adapt across domains. By combining the two, our approach demonstrates portability, reproducibility, and efficiency, while producing quality scores that reveal a clear trajectory for improvement. In financial and biomedical domains, better relation extraction and tighter integration with entity resolution could increase graph connectivity. In document-level tasks such as DocRED, augmenting rule-based extraction with lightweight LLM-based classification may correct systematic misclassification errors. In all cases, the feedback loop provided by the LLM-as-a-Judge can serve as a foundation for semi-automatic refinement of extraction rules.

Overall, the experiments confirm that end-to-end knowledge graph construction and validation can be achieved in a cost-effective and scalable manner across domains. The framework succeeds in producing domain-specific graphs with varying degrees of precision and connectivity, while the validation stage ensures that errors are not only measured but also interpreted. This positions the pipeline as both a practical system for applied KG construction and a methodological contribution for research in hybrid approaches to information extraction.

# 6   Conclusion and Future Work

This paper introduced a hybrid framework for the automated construction and validation of knowledge graphs, combining LLM-driven ontology induction with scalable rule-based methods for extraction, resolution, and assembly. Evaluations across three benchmark datasets — FiQA in the financial domain, DocRED for document-level relation extraction, and CDR in the biomedical domain — demonstrate both the versatility and the limitations of this approach. In all cases, the pipeline produced structured knowledge graphs at minimal cost, with total expenses ranging from only $0.15 for DocRED to under $3 for FiQA. Processing times were consistently short, measured in minutes rather than hours, confirming the efficiency of the system.

The experiments revealed complementary strengths and weaknesses across domains. In FiQA, ontology induction successfully captured a rich conceptual schema, but relation connectivity was sparse and entity classification errors persisted. In DocRED, dense annotations exposed the limits of purely rule-based extraction, with entity type misclassification dominating the error profile. In contrast, the biomedical CDR dataset showed that domain-specific ontology induction and tailored extraction rules could yield high semantic quality, even if graph-level sparsity remained a challenge. The LLM-as-a-Judge validation framework proved invaluable across all settings, not only quantifying node, edge, and graph quality, but also surfacing systematic issues such as vague entity labels,

semantic inconsistencies, or structural fragmentation. These results validate the design choice of combining LLM reasoning with rule-based scalability, producing actionable insights at low cost.

Future work will build on these findings in three directions. First, improvements in entity classification and relation extraction are necessary to enhance precision and reduce sparsity, particularly in financial and general knowledge corpora. This may include expanding the pattern libraries, integrating contextual cues, and leveraging lightweight LLM components for disambiguation. Second, tighter integration between entity resolution and relation extraction should preserve more valid connections, increasing graph connectivity without inflating noise. Third, we plan to extend the role of LLMs beyond ontology induction and validation to include semi-automatic refinement of extraction rules and support for entity canonicalization. Additional avenues include multi-model validation, cross-domain adaptation beyond the three benchmarks studied here, and the development of new quality metrics that capture not only correctness but also coverage and coherence.

In conclusion, the presented framework establishes a cost-effective, modular, and adaptable pipeline for knowledge graph construction. By demonstrating its applicability across financial, general, and biomedical domains, and by coupling construction with explicit validation, we provide both a practical methodology and a foundation for future research. The system offers a clear path toward richer, more connected, and more accurate knowledge graphs, while remaining accessible in terms of computational cost and reproducibility.

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
