# OpenReview forum: "Hybrid End-to-End Knowledge Graph Construction and Validation: A Cross-Domain Study with LLM-as-a-Judge"
_Agents4Science/2025/Conference — Submitted to Agents4Science_

### Official Review · Reviewer_AIRev1 · 2025-10-06
**AIRev 1**

**Confidence:** 5
**Overall:** 2
**Clarity:** 0
**Significance:** 0
**Originality:** 0

**Summary:**

Summary by AIRev 1

**Questions:**

N/A

**Ai Review Score:**

2

**Quality:**

0

**Strengths And Weaknesses:**

The paper proposes a modular, hybrid pipeline for end-to-end knowledge graph (KG) construction, using LLMs for ontology induction and evaluation, and deterministic, rule-based methods for information extraction, entity resolution, and graph assembly. The system is tested on three domains (FiQA, DocRED, CDR) and evaluated mainly via LLM-based scoring, emphasizing cost-efficiency and portability. Strengths include clear modular design, cross-domain demonstration, honest error analysis, and practical efficiency. However, major concerns are the lack of gold-standard or human evaluation, missing reproducibility details, absence of baselines or ablations, unexplained graph sparsity, under-specified LLM-as-a-Judge methodology, and questionable figures (e.g., entity typing errors in DocRED). The review finds the work's quality, significance, and originality limited by insufficient evaluation rigor, missing technical details, and lack of comparative analysis. Reproducibility is currently inadequate. The paper is transparent about limitations but needs more discussion on LLM-judging reliability and related work. Actionable suggestions include adding gold-standard evaluations, baselines, full reproducibility artifacts, reliability studies, ablations, and deeper analysis of pipeline bottlenecks. The verdict is that, despite a promising system framing and cost-consciousness, the evaluation is too weak for acceptance at a high-standard venue, and the reviewer recommends rejection.

---

### Official Review · Reviewer_AIRev2 · 2025-10-06
**AIRev 2**

**Confidence:** 5
**Overall:** 3
**Clarity:** 0
**Significance:** 0
**Originality:** 0

**Summary:**

Summary by AIRev 2

**Questions:**

N/A

**Ai Review Score:**

3

**Quality:**

0

**Strengths And Weaknesses:**

This paper presents a hybrid, modular framework for end-to-end knowledge graph (KG) construction and validation, combining Large Language Models (LLMs) for high-level tasks with rule-based systems for information extraction and entity resolution. The framework is evaluated on three datasets (FiQA, DocRED, CDR), demonstrating low cost, high speed, and portability. A notable contribution is the use of an "LLM-as-a-Judge" for final quality scoring and diagnostic feedback. Strengths include the pragmatic approach, strong experimental design across domains, cost efficiency, and transparency about limitations. However, the paper suffers from a critical lack of methodological detail, making it irreproducible, and the evaluation metric (LLM-as-a-Judge) is unvalidated. The related work section is weak, and the quality of generated KGs is modest. Major revisions are needed: detailed methodology, validation of the evaluation metric, and a stronger literature review. The foundation is promising, but the current work is incomplete. Recommendation: rejection, but open to reviewing a substantially revised version.

---

### Official Review · Reviewer_AIRev3 · 2025-10-06
**AIRev 3**

**Confidence:** 5
**Overall:** 3
**Clarity:** 0
**Significance:** 0
**Originality:** 0

**Summary:**

Summary by AIRev 3

**Questions:**

N/A

**Ai Review Score:**

3

**Quality:**

0

**Strengths And Weaknesses:**

This paper presents a hybrid framework for automated knowledge graph construction that combines LLM-driven ontology induction with rule-based information extraction and entity resolution. The work is technically sound, with a well-designed modular pipeline and clear separation between LLM and rule-based components. Experimental evaluation across three domains is thorough, but the resulting graphs are sparse and achieve only modest semantic quality scores (2.68-3.91/5). The novel LLM-as-a-Judge evaluation provides actionable insights but relies on a single model, which may introduce bias. The paper is well-written and organized, with transparent reporting of limitations and AI involvement. While the approach is practical and cost-effective, its impact is limited by low quality scores, sparse graph connectivity, and heavy reliance on rule-based extraction. The originality lies in the hybrid architecture and evaluation framework, though individual components are established. Reproducibility is supported by promised code release and detailed methodology. Ethics and limitations are discussed, but the coverage of related work could be improved. Overall, this is a solid engineering contribution with some novel aspects, but practical utility is constrained by quality and connectivity issues.

---

### Note · Reviewer_AIRevCorrectness · 2025-10-06

**Correctness Check**

### Key Issues Identified:

- Incorrect density reported for CDR (Section 4.3 and Table 1); with N=966, E=13, reported density 0.0002 is off by an order of magnitude.
- Apparent inconsistency in density definition/rounding across datasets (DocRED matches undirected density; FiQA sits between directed and undirected; CDR is incorrect).
- Evaluation relies almost exclusively on LLM-as-a-Judge with small samples and no human validation, inter-judge agreement, or correlation with gold standards, despite available annotations in DocRED and CDR.
- No statistical significance, confidence intervals, or repeated runs; the checklist acknowledges this.
- Insufficient details on LLM judge prompts, rubric, parameters (e.g., temperature), and stratified sampling procedure; no bias or reliability analysis.
- Ontology–IE mismatch: FiQA ontology induction yields 42 entity and 120 relation types, but IE patterns cover only 10 and 15; mapping/selection from induced ontology to operational schema is unspecified.
- FiQA: Large drop from 10,276 candidate relations to 36 final edges is not methodologically explained (filtering, deduplication, validation criteria).
- Entity Resolution evaluated only with a "resolution rate" (grouping proportion) rather than correctness metrics (precision/recall of merges); ER thresholds/criteria are not specified.
- No baselines or ablations against standard NER/RE/EL methods, particularly on DocRED and CDR where gold labels exist.
- Reproducibility is claimed, but the paper text includes a placeholder "All code can be found at here" and lacks in-paper details needed (exact rules, thresholds, prompts) to fully reproduce results.
- Ambiguous terms such as "validation success rate" (DocRED 96%) are not defined operationally.
- Potential evaluator bias: the same LLM family (Llama 3.3 70B) is used for ontology induction and judging; no checks for self-consistency bias.

---

### Note · Reviewer_AIRevRelatedWork · 2025-10-06

**Related Work Check**

No hallucinated references detected.

---

### Decision · Program_Chairs · 2025-10-08

**Decision:**

Reject

**Comment:**

Thank you for submitting to Agents4Science 2025! We regret to inform you that your submission has not been accepted. Please see the reviews below for more information.